# Monitoring of Evolving Laser Induced Periodic Surface Structures

**Andrea Lübcke [1] Zsuzsanna Pápa [2,3] and Matthias Schnürer [1,*]**

[1] Max-Born-Institut für nichtlineare Optik und Kurzzeitspektroskopie, Max-Born-Strasse 2a, 12489 Berlin, Germany

[2] MTA "Lendület" Ultrafast Nanooptics Group, Wigner Research Centre for Physics, 1121 Budapest, Hungary

[3] ELI-ALPS Research Institute, ELI-HU Non-Profit Ltd., 6720 Szeged, Hungary

[*] Correspondence: schnuerer@mbi-berlin.de

**Abstract:** Laser induced periodic surface structures (LIPSS) are generated on titanium and silicon nitride surfaces by multiple femtosecond laser pulses. An optical imaging system is used to observe the backscattered light during the patterning process. A characteristic fringe pattern in the backscattered light is observed and evidences the surface modification. Experiments are complemented by finite difference time domain numerical simulations which clearly show that the periodic surface modulation leads to characteristic modulations in the coherently scattered light field. It is proposed that these characteristic fringe pattern can be used as a very fast and low-cost monitor of LIPSS formation formation during the manufacturing process.

**Keywords:** laser induced periodic surface structures; laser nanostructuring; material processing; optical properties

## 1. Introduction

Laser induced periodic surface structures (LIPSS) have been in the focus of surface functionalization for a many years. They have been observed for the first time by Birnbaum in 1965 [1] and can be generated by linearly polarized laser pulses on almost all material classes, for example, metals [2–5], semiconductors [6–8], dielectrics [9–11].

LIPSS are easy to fabricate with femtosecond laser systems of moderate power and have found widely spread application (cf. e.g., Reference [12] and references therein) influencing the physical and chemical properties of surfaces. They can be applied to control wettability and friction of surfaces [13,14], to generate structural colors [15] or to optimize cell growth in biomedical applications [16].

There are two different LIPSS types: Low Spatial Frequency LIPSS (LSFL) with periods on the order of the laser wavelength and High Spatial Frequency LIPSS (HSFL) with periods significantly shorter than the laser wavelength. The orientation of the LIPSS depends on their type and the material. In metals, LSFL are oriented perpendicular to the laser polarization.

The formation of LIPSS in metals by ns pulses can be described by the interference of the incident laser beam with a surface-scattered electromagnetic wave [17–19] leading to inhomogeneous energy deposition into the material. The formalism has been extended towards fs-pulse interaction with metal surfaces by Bonse and coworkers [9,20]. There is no consensus yet on the underlying principles leading to HSFL. Different models have been discussed covering nonlinear frequency conversion [9,21], nanoplasmonic excitation [22] and self-organization [23].

In order to study the formation process and to gain information on the underlying principles, different monitoring processes have been developed, for example, in situ diffraction [24], in situ

imaging deploying a microscope [25] and in situ surface second harmonic generation [26]. In situ imaging and diffraction requires a pump probe setup, that is, a probe beam at lower wavelength in addition to the patterning beam.

Here, we present another monitoring method which is based on an interesting discovery in our recent experimental work. In that work, LIPSS on titanium and $Si_3N_4$ samples were fabricated in-situ as targets for laser particle acceleration and X-ray generation [27,28]. In such experiments laser pulses with ultra-high intensity $\sim(10^{18} \ldots 10^{20})$ W/cm$^2$ are applied to a solid target, where ions from a contamination layer at the target rear side are accelerated due to the target normal sheath acceleration (TNSA) mechanism [29]. Depending on the target material, an X-ray flash is released in addition as result of the high intensity laser target interaction [30]. LIPSS have been generated by the same but strongly attenuated laser pulses. As soon as LIPSS have been formed, a characteristic fringe pattern in the back-reflected light is observed which is the basis of the proposed imaging system. The method proposed in the present work does not require an additional monitoring beam but analyzes the backscattered light of the patterning laser.

Currently, much effort is spent to advance such secondary radiation sources towards repetitive applications. LIPSS formation is a very promising target manufacturing method capable to provide optimized laser-target interaction and—given the availability of a monitoring system—compatible with high repetition rate applications.

Such simple monitoring system will not only be essential in facilitating efficient high repetition rate laser driven secondary radiation sources but will be helpful in any application that needs very fast evaluation of LIPSS formation processes (ideally while writing the structures), such as required for inspection of large surface in industrial applications [31].

In this work we investigate the origin of the fringe pattern with the help of of a 3D Maxwell solver based on the finite difference time domain (FDTD) method. The modulation period found in simulation accounts for the fringe pattern observed in experiment. After description of the experimental setup, parameters and optical imaging system we display produced surface structures and images of the scattered light fields. Details of the simulations are briefly summarized and different case calculations are discussed to underpin that LIPSS are the origin of the observed fringe pattern.

## 2. Experimental

Experiments have been performed at the Max-Born-Institute High Field Ti:Sapphire laser. In the current work we investigated nanostructuring of 1 μm thick titanium and $Si_3N_4$ foils within a proton acceleration experiment. Details for the laser acceleration experiments are described elsewhere [27]. Here, we concentrate only on the nanostructuring part. The pulse duration of the p-polarized laser is about (30–35) fs. The experimental setup is shown in Figure 1.

After carefully adjusting the target position, such that the focal plane of the off-axis parabolic mirror (OAP) lies in the target plane (focal distance = 15 cm), an aperture is introduced into the beam path and reduces the beam diameter to ~6 mm. The energy on target was carefully measured and could be varied by changing a laser amplifier delay. Typically, we used energies of about 10–30 μJ. With the reduction of beam diameter, the focal spot size increases to FWHM of about 25 μm and the spatial profile shows the typical Airy pattern. Under these conditions we were able to generate LIPSS in the central part of the focal spot. The intensity in the Airy-rings was not sufficiently high in order to induce ripple formation. LIPSS were characterized both by scanning electron microscopy (SEM) and atomic force microscopy (AFM). Typical results are shown in Figure 2. SEM and AFM also revealed that the initial surface quality of the samples differs significantly. While $Si_3N_4$ has a nearly perfect optical surface, the titanium foils are relatively rough. In agreement with others, LIPSS in Ti have a periodicity of ~600–800 nm, a height of ~200 nm [3,32] and an orientation perpendicular to the polarization vector of the laser field. However, in contrast to those works, relatively low pulse numbers (5–50) and relatively high fluences (2–6 J/cm$^2$) were applied. Furthermore, our laser pulse duration was shorter and the experiments were performed in vacuum. LIPSS on $Si_3N_4$ have a lower

periodicity of ∼250 nm and a lower height of about 50 nm but the same orientation with respect to the laser field polarization vector. Occasionally, we have also observed larger structures in $Si_3N_4$ but those structures are usually restricted to a small part of the structured region. Here, the periodicity is again about 800 nm and the height is about 140 nm. Our setup comprises a backreflection imaging system: Backscattered light is collected and collimated by the same off-axis parabolic mirror that focuses the laser pulse onto the target. Part of this backscattered light leaks through a dielectric mirror (M2 in Figure 1) and is refocused onto a CCD camera (PIXIS with $(25 \, \mu m)^2$ quadratic pixels) by a lens (focal distance of 150 cm). Thus, the imaging system has a magnification of 10×. Mirror M2 may also be replaced by a beam splitter. In our particular setup the mirror was required to deliver highest possible pulse energies to the sample for the laser acceleration experiments.

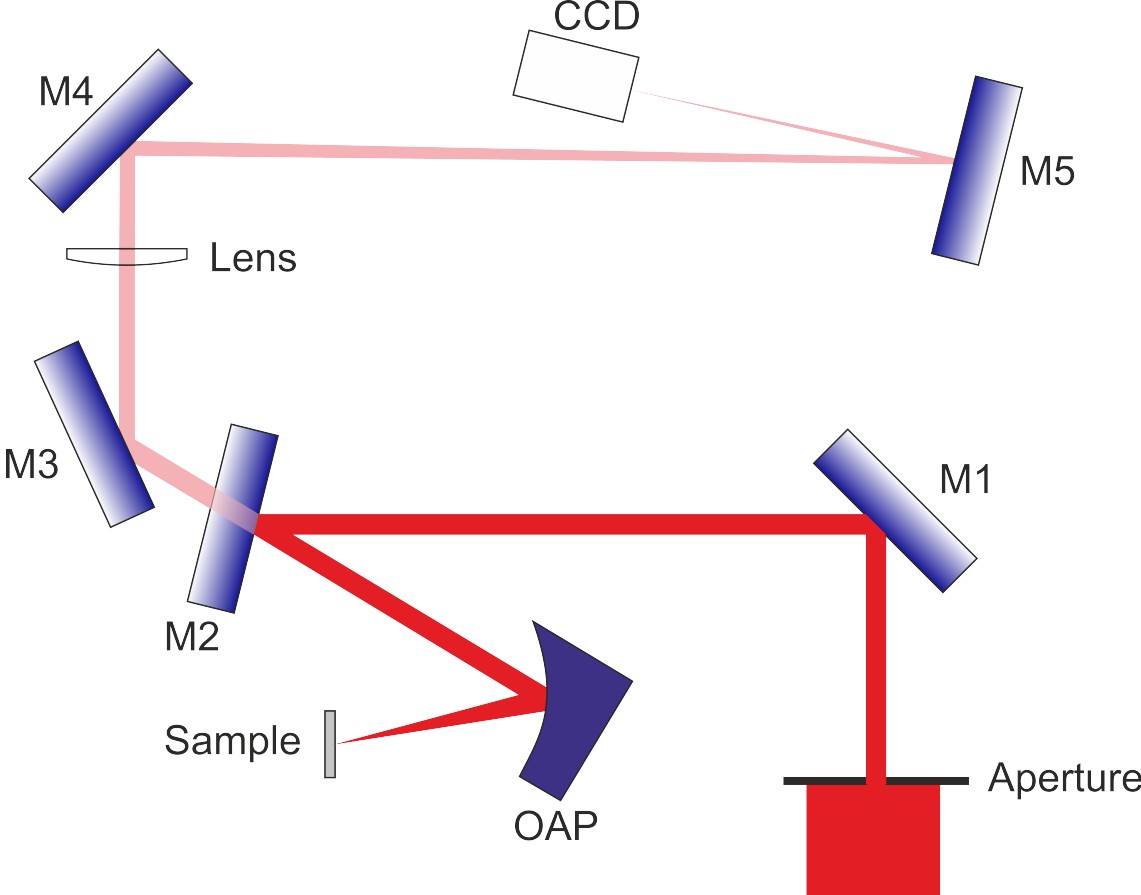

**Figure 1.** Experimental setup. Backscattered light from the target is collected by an off-axis parabola (OAP). Part of it is leaking through a mirror (M2) and is focused by a lens onto a CCD camera.

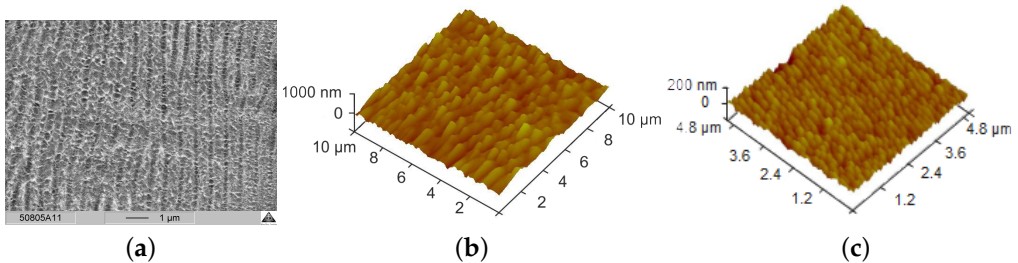

**Figure 2.** SEM (**a**) and AFM images of nanostructured titanium foil (**b**) and AFM image of nanostructured $Si_3N_4$ (**c**).

With the onset of LIPSS generation, our backreflection changes dramatically: a secondary diffraction pattern (in addition to the Airy pattern due to the aperture) appears with LIPSS (cf. Figure 3). The fringe distance is 3–4 px, corresponding to 75–100 µm in the image plane. We do not observe a significant difference between the fringe patterns for titanium or $Si_3N_4$ samples.

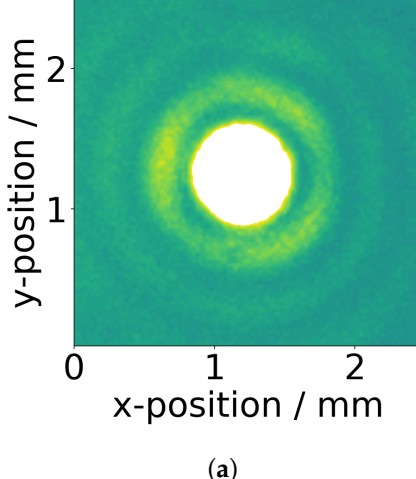

(**a**)

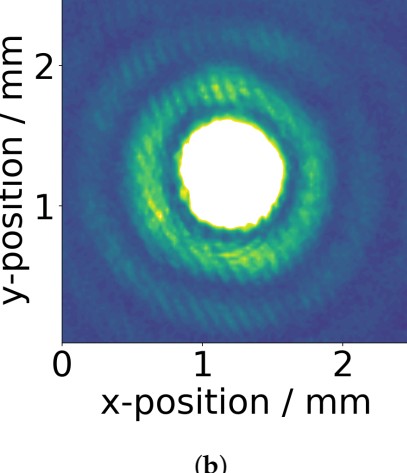

(**b**)

**Figure 3.** $Si_3N_4$ sample surface imaged during nanostructuring by the back reflection diagnostic. (**a**) first pulse, (**b**) back reflection after the structure has been formed. Scales refer to the image plane. Magnification factor: $M = 10\times$.

## 3. Computational

In order to investigate where the observed secondary fringe pattern is originating from, we carried out numerical simulations with Lumerical FDTD Solutions. Finite difference time domain (FDTD) method is widely applied for solving different electromagnetic problems by discretizing Maxwell's equations, both in time and space with central difference approximations. Typically a Cartesian volume element having $\Delta x$, $\Delta y$, $\Delta z$ dimensions is used for the space discretization (the most common discretization technique is the Yee scheme [33,34]) and a time step $\Delta t$ for the time discretization [35]. In the FDTD method electromagnetic radiation (e.g., a plane wave) is injected to the simulation volume at some initial time and new field components are computed from differences based on the field components of the previous time interval. This process is continued iteratively until the transient solution for the fields has converged to a steady-state solution. The near-field solution that one obtains for the fields inside the computational domain is then Fourier transformed into the frequency domain and subsequently propagated into the far-field. This computational approach is extensively used for studying scattering problems of nanostructured surfaces [36].

Our goal was to model the far-field response of the nanostructured surface illuminated by laser pulses. We applied a linearly polarized light source with central wavelength of 800 nm, with Airy-disk beam profile having 25 µm FWHM diameter. The simulation volume had dimensions of $60 \times 60 \times 1.2\ \mu m^3$. Perfectly matched layer boundary conditions were used on all sides of the calculation domain. The model system consisted of a substrate with a structured central part of 30 µm diameter in the same geometrical arrangement as for the laser irradiated sample. The nanostructure is modeled by a one-dimensional sinusoidal height modulation. The direction of the periodic structures was perpendicular to the polarization direction of the illumination. We applied two substrates, Ti and $Si_3N_4$. Their dielectric properties were taken from references [37,38]. The whole structure was surrounded by vacuum (index of refraction $n = 1$). To ensure an accurate calculation of the reflected beam from the surface structures, mesh/grid size of 24 nm was chosen. A 2D frequency domain field profile monitor at 500 nm above the surface was used to visualize the field distribution of the

reflected light components. The data of this monitor was later applied to calculate far-field response of the structured surface by letting the field components propagate to a given distance in space. This way, we could calculate the static response of the structured sample surface in the far-field upon laser illumination.

## 4. Results

Figure 4 shows the simulated two-dimensional near field distribution of the light backreflected from a structured titanium sample. The structure is applied only in the central region with a diameter of 30 μm.

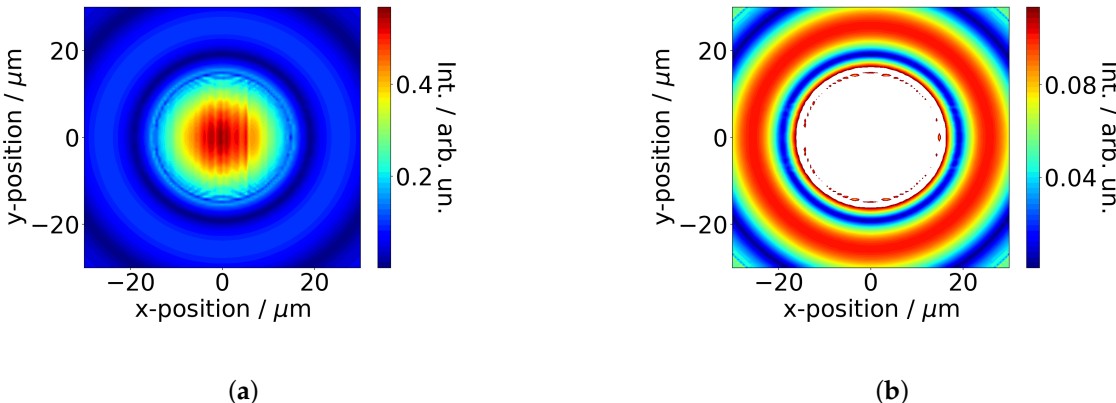

(**a**)　　　　　　　　　　　　　　　　　(**b**)

**Figure 4.** Simulated near field distribution of the reflected light at 500 nm distance from the surface. Two different color scales are used to show that fringes are visible only in the central part (**a**) and not in the Airy ring (**b**).

The near field distribution shows modulations in the central region of the reflected beam (where the incident beam interacted with the structured region) (Figure 4a) but not in the Airy rings (cf. Figure 4b).

In Figure 5 we show the simulated far-field distribution of the backreflected light from (a) an unstructured and (b) a structured titanium surface at 50 μm distance from the sample surface. In this case, the structure was applied to the entire sample, the period of the structure was 600 nm and its height was 300 nm.

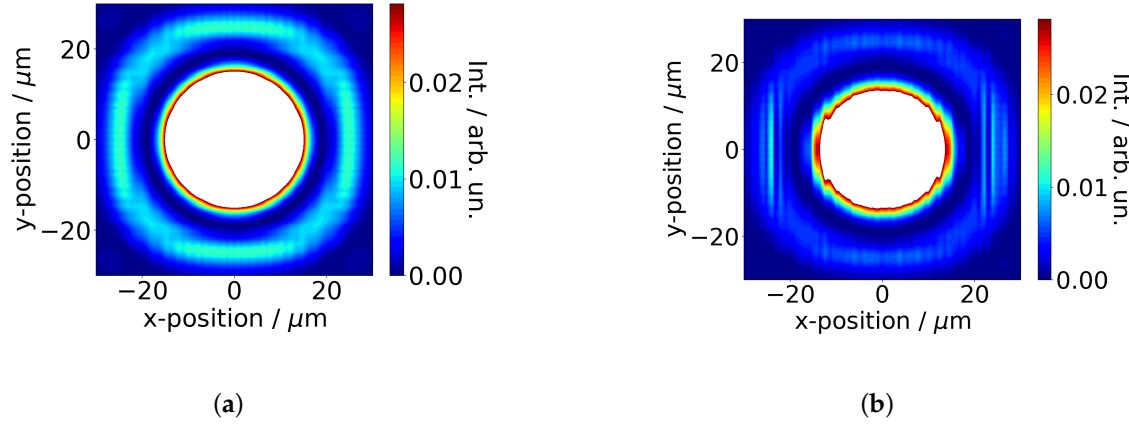

(**a**)　　　　　　　　　　　　　　　　　(**b**)

**Figure 5.** Simulated far-field distribution of the reflected light at 50 μm distance from the surface. (**a**) Unstructured titanium. (**b**) Sinusoidally structured titanium surface (600 nm period, 300 nm height).

The color scale was chosen such that modifications in the Airy rings are clearly visible, that is, the central spot is saturated.

The intensity of the light backscattered from the structured sample is strongly reduced. This is because the sub-wavelength nanostructure strongly reduces the reflectivity of the sample. In addition, we clearly observe the appearance of vertical fringes in the presence of the nanostructure. In order to elucidate the origin of the observed fringe pattern, we will numerically investigate the role of relevant parameters, in the following.

### 4.1. Influence of Geometrical Properties of the Surface Structure

### 4.1.1. Size of Structured Area

We first tested if the fringes appear in the simulation allover the beam profile even if only the central part of the sample is structured. This is important since in the experiment fringes are observed in the Airy rings although the structure is created only in the central beam part. The results of comparative simulations are shown in Figure 6. To emphasize the structure-induced changes to the far-field we plot the difference between the far field of the light backreflected from the structured and unstructured surface. In (a) we show the structure-induced difference, if the structure is applied all over the sample while in (b) the structure is present only in the central 30 μm diameter spot. As already seen in Figure 5, the entirely structured sample shows a reduced reflectivity and the appearance of vertical fringes. If only the central part is structured (b), the reflectivity in the region of the Airy rings is nearly unchanged but in addition to the vertical fringe pattern also a circular fringe pattern appears which originates from diffraction from the circularly shaped structured area. The distance between the circular fringes is significantly larger than the period of the vertical fringe pattern. From these simulations we conclude that the fringes observed in the Airy rings are caused by the structure in the central part of the beam.

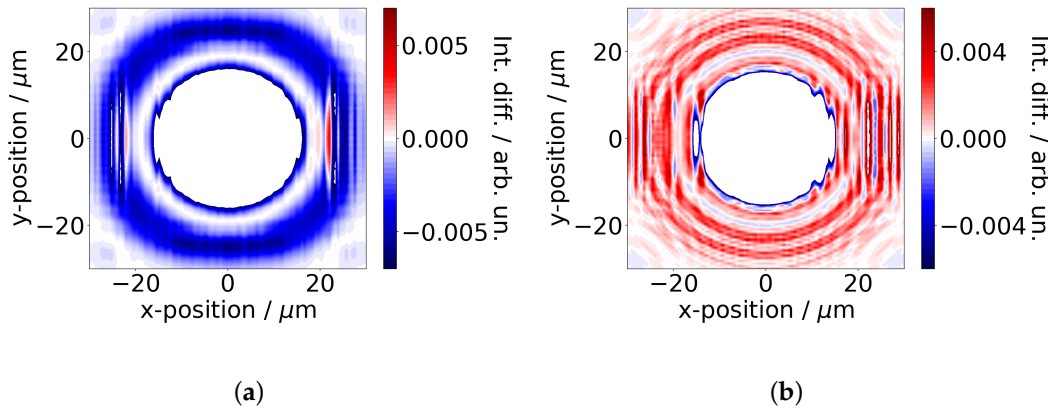

(**a**)　　　　　　　　　　　　　　　　　　　　　　　　　(**b**)

**Figure 6.** Structure-induced intensity changes to the far-field distribution at 50 μm distance, that is, the difference between the far-field distribution of reflected light from structured and unstructured samples. (**a**) For an entirely structured sample surface. (**b**) For a sample where only the central spot (30 μm diameter) is structured.

### 4.1.2. Influence of Structure Period

As described in the experimental section, similar fringe patterns were observed for titanium and $Si_3N_4$ samples although the structural parameters were very different as well as the dielectric properties of the samples. Therefore, we have investigated the role of the structure period for the 2D electric field distribution. The resulting fringe pattern only slightly depends on the structure period: Fourier analysis of the field distribution yields fringe periods of ∼1.55 μm, 2.01 μm, 2.17 μm and 2.35 μm for nanostructured titanium samples with a structure height of 300 nm and structure periods of

200 nm, 600 nm, 700 nm and 800 nm, respectively. The highest contrast of the fringe pattern is observed for the 800 nm structure.

### 4.1.3. Role of Crater

In the case of the titanium samples, during target structuring also a crater was formed. In contrast, the structured $Si_3N_4$ samples do not show a crater. In principle, it could be conceivable that a fringe pattern is also generated by interference of the light reflected from an (unstructured) crater ground and light reflected from the unmodified surface. Our simulations show that the circular crater generates an additional circular diffraction pattern. But only if the ripple surface structure is present also the vertical fringe pattern appears. We observe a small variation of fringe pattern with the crater depth: The observed fringe period slightly decreases with increasing crater depth. While a 600 nm structure without crater shows a fringe pattern with 2.0 µm period and a 250 nm deep crater almost leaves the fringe period unchanged, a 500 nm deep crater reduces the fringe period to 1.8 µm. We also investigated, whether the different distance of the surface structure from the detector (due to the presence of the crater) can account for the change of fringe period but did not find any change on these scales.

### *4.2. Role of Optical versus Geometric Properties*

So far, we used the optical properties of cold titanium surface for all simulations. However, the experimental results show that very similar fringe patterns are observed for both the titanium and $Si_3N_4$ samples. This is particularly interesting since not only the structures in both cases are significantly different but also the optical properties, that is, one question that needs to be addressed is the role of the carrier density. It is indeed conceivable that the increase of carrier density during the interaction with the laser pulse is essential for the generation of the fringe pattern as this modifies the optical properties significantly. $Si_3N_4$ has a wide bandgap ($\sim$4.5 eV), that is, at least the absorption of three photons is required to generate an electron hole pair. Nevertheless with the applied fluences and intensities a significant increase of carrier density during the laser solid interaction must be expected. We have compared the structure-induced changes for a titanium and a $Si_3N_4$ sample with the same structural parameters (200 nm structure period, 50 nm structure height). Fringe periods are 1.56 µm and 1.28 µm for titanium and $Si_3N_4$.

The far-field strength is significantly higher in case of the metal surface. While the circular fringe pattern originating from diffraction from the circular structured shape is clearly visible in both cases, the vertical fringe pattern is only present in case of a metal, that is, for high carrier density. Thus, to become sensitive to LIPSS formation, a high carrier density is needed in the instant at which the backscattered signal is generated. that is, backscattered light from a cold insulator (low carrier density) will not carry sufficient information about the presence of the structure. Analyzing the backscattered light from the pulse generating the structure is a very simple approach to gain information about the structure presence.

The results of these simulations are supported by experimental observations: We did not succeed to experimentally reproduce the fringe pattern in the backreflected light when illuminating LIPSS at much lower intensities.

## 5. From Fringe Patterns in the Far Field to the Experimentally Observed Fringes

So far, we have investigated how the far field distribution at a given distance from the sample surface is affected by the presence of a surface nanostructure. This does not directly relate to the experimental observation of the fringe pattern since an imaging system is involved. To further elucidate the origin of the fringe pattern, we investigate the far-field distribution as function of distance from the sample surface. Figure 7 shows a two-dimensional map of the far field distribution $E^2(x, y = 0, z)$. The color scale is chosen such that the modulations in the Airy rings are easily visible, that is, the central part of the beam profile is saturated.

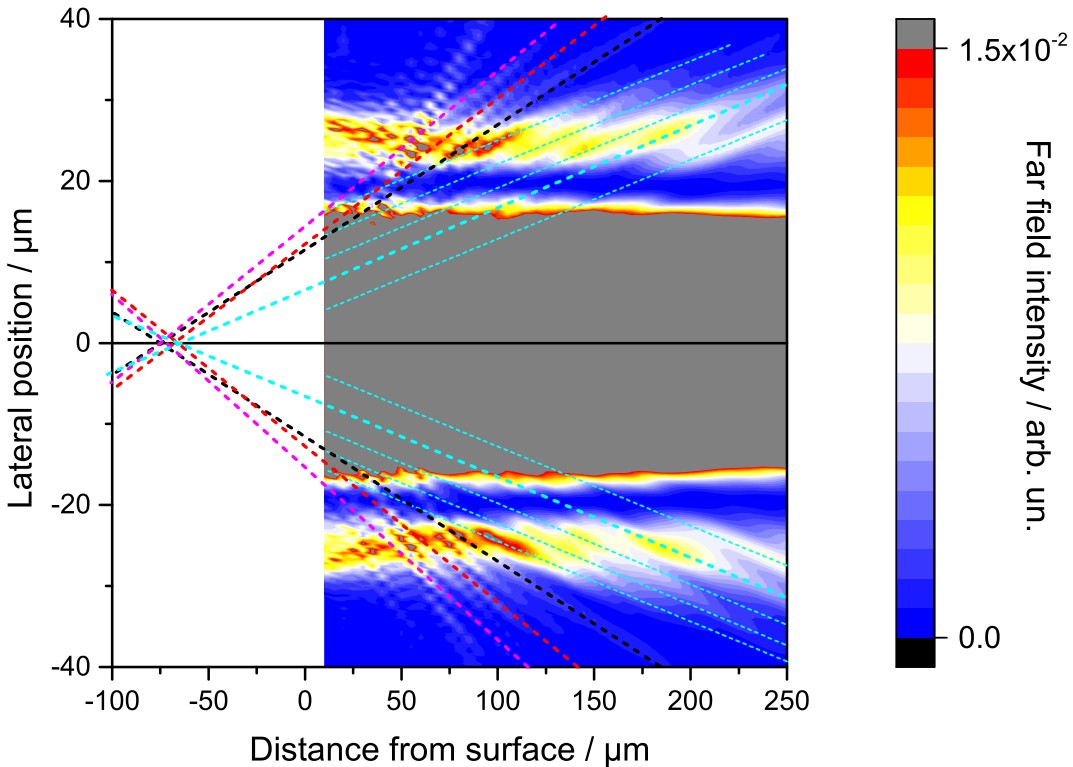

**Figure 7.** Simulated far-field distribution map $E^2(x, y = 0, z)$ as function of distance from the sample surface which is located at $z = 0$.

The 2D-map shows that the modulations already observed in the farfield distribution at a given distance actually originate from new, discrete angular contributions to the outgoing wave. Some examples are highlighted by thick, coloured lines. These new angular components are generated by the surface nanostructure. It seems that they originate from a virtual source around $(65\dots75)$ µm behind the sample. The angles with respect to the sample normal are $\alpha = \sim5°$ (cyan), $\sim9°$ (black), $\sim10°$ (red) and $\sim11°$. In addition, we not only observe single beams outgoing at different angles but bundles of beams as exemplarily shown for the cyan lines. Those contributions outgoing under sufficiently small angles can be imaged by our imaging system. The (half) acceptance angle of the parabola is $14°$, that is, the highligted beams will be transported by the imaging system. Since the virtual source lies behind the sample plane, these newly generated discrete beams will be focused in front of the original image plane. This is sketched in Figure 8. The original beam from the unstructured surface(red) is focused in the image plane and exhibits an intensity profile similar to the beam incident on the sample surface with the Airy diffraction pattern of the aperture(thick red curve). Since the reflectivity in the structured region is strongly reduced, the intensity ratio between the individual intensity maxima of the reflected light may however be different from the incident light. In the image plane, the wave vector (thick, red arrow) of this beam is parallel to the optical axis. The blue line shows the LIPSS generated beam with its focal plane in front of the original image plane, that is, its wave vector (thick blue arrow) includes a finite angle $\beta$ with the original beam (reflected from the surface) which leads to interference in the image plane.

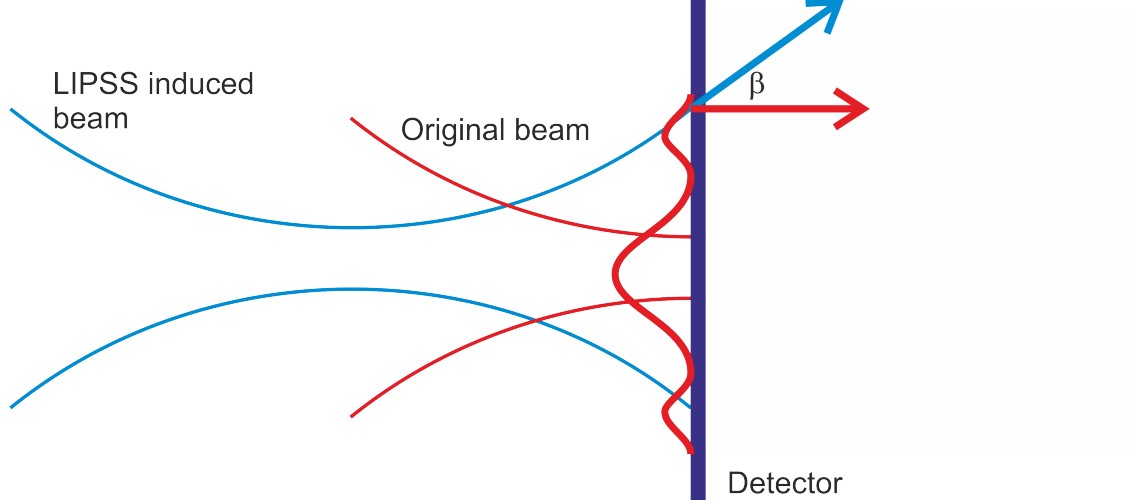

**Figure 8.** New LIPSS-induced angular components to the backscattered light are focused in front of the original image plane. Interference between the original and LIPSS-induced beam in the detector plane gives rise to the experimentally observed fringe pattern.

$\beta$ is given by $\alpha$ and the magnification factor of the imaging system $M = 10$ via $M \tan \beta = \tan \alpha$, that is, for new angular components of $\sim 5°$ on the sample side, the intersection angle of the two wavefronts at the image plane is $\sim 0.5°$. Interference between these two tilted beams give rise to fringes with a distance of $\Delta L = \frac{\lambda}{\sin \beta} \sim 90\,\mu\text{m}$, which is in very good agreement with our experimental observation of $(75 \ldots 100)\,\mu\text{m}$.

So far, we have concentrated on the smallest angles observed in the farfield distribution. The larger angles will lead to smaller fringe distances, for example, for the $9°$ angle on the sample side, we expect fringe periods of $50\,\mu\text{m}$ in the image plane, which corresponds to only 2 pixels of our CCD camera. This is the Nyquist limit to reproduce the fringe pattern. Given the assumption made in the numerical model and the precision in determining the angular contributions it is conceivable that those fringe patterns are not resolved with our setup.

## 6. Conclusions

In this work we have described an optical setup which is capable to monitor the appearance of laser induced periodic surface structures. In particular, we have shown that the generation of LIPSS on titanium and $Si_3N_4$ surfaces give rise to a characteristic fringe pattern in the backscattered light of the patterning laser. This fringe pattern was experimentally observed and very well reproduced by Lumerical simulations using finite difference time domain method. They clearly show that the fringe pattern is unambiguously related to the periodic surface structure. Nevertheless, some questions remain open. We could not yet unambiguously resolve the role of carrier density. And also the fact that the very different LIPSS periods in $Si_3N_4$ and titanium lead to quite similar fringe pattern is surprising. The described backreflection diagnostics is therefore a very simple way to monitor the generation of LIPSS with sub- laser wavelength periodicity *during* the structuring process. This on-line monitoring technique will allow fabricating and using nanostructured targets/samples within a single experimental step, that is, without time-demanding sample characterization. One very promising application could for instance be the use of nanostructured targets for high-repetition rate laser ion acceleration (e.g., in radiation therapy): ions are accelerated by laser pulses (e.g., at 10 Hz) making use of tape targets. In between the individual high energy laser pulses, a high repetition rate laser can structure the sample. Its backreflection gives information about the presence of the structure.



**Author Contributions:** Conceptualization, A.L., Z.P. and M.S.; methodology, A.L., Z.P. and M.S.; validation, A.L., Z.P. and M.S.; formal analysis, A.L. and Z.P.; investigation, A.L., Z.P. and M.S.; writing—original draft preparation, A.L.; writing—review and editing, A.L., Z.P. and M.S.; visualization, A.L. and Z.P.; funding acquisition, Z.P. and M.S.

**Funding:** This research was funded by Deutsche Forschungsgemeinschaft (project—Relativistic Nano-Plasma Photonic SCHN953/2-1) and National Office for Research, Development and Innovation (FK 128077).

**Acknowledgments:** We thank C. Lienau (University Oldenburg), R. Grunwald (MBI), B. Pfau (MBI) and J. Bonse (BAM) for extensive and fruitful discussion. In particular, we thank C. Lienau for triggering our collaboration and J. Bonse for providing additional samples. MS acknowledges funding from the Deutsche Forschungsgemeinschaft (project—Relativistic Nano-Plasma Photonic SCHN953/2-1). ZP thanks the support from National Office for Research, Development and Innovation (FK 128077).

**Conflicts of Interest:** The authors declare no conflict of interest. The funders had no role in the design of the study; in the collection, analyses, or interpretation of data; in the writing of the manuscript, or in the decision to publish the results.

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
