# Peer review of "Monitoring of Evolving Laser Induced Periodic Surface Structures"

_applsci, doi:10.3390/app9173636_

Round 1

Reviewer 1 Report

This paper shows an imaging system inspired by an experiment previously performed by the authors in which a characteristic fringe pattern in the reflected light was observed on titanium and Si3N4 samples. This experimental paper is well organized, and both the experiments and results are clearly exposed.

Some minor issues to be considered:

Both the abstract and the conclusions can be more specific about the work. On one hand, the abstract can mention the way the simulations were preformed and summarize the results; on the other hand, the conclusions could include an introduction about the what is presented in the paper. Some open questions remain, and they should be highlighted in the document: the reasons why there is no difference in fringe patterns for titanium or Si3N4, how the different distance of the surface from the detector can affect the fringe period, the role of the carrier density, ... I miss information about the computational details and the performance of the simulations. More references could help the reader that is not directly related to the field.

Reviewer 2 Report

The authors observed periodic surface structures using a 35 fs Ti:Sapphire laser. The method is based on the authors previous experimental work, namely references 4 and 5. The interaction of fs laser and periodic structure surfaces is explained. The Si3N4 is a 200 nm period and 50 nm height structure. Fringe periods are 1.56 and 1.28 microns. These numbers are comparable to the wavelength of 800 nm incident beam. The manuscript is well organized, should be published after a minor revision. The authors mentioned about one very promising application as "nanostructured targets for high-repetition rate laser ion acceleration". The ion acceleration is, however, not quite related to the main topic of observation. Also, the description is very limited. It is suggested that the author enrich the description here for general readers, and provide one more possible application. This technique can be useful for fast surface characterizations and fundamental research: 1. "A very simple, fast and low cost method was used to evaluate the effectiveness of the LIPSS generation, capable of examining a large surface area as required in industrial applications." [Surface and Coatings Technology, Volume 344, 25 June 2018, Pages 423-432] 2. The technique can be quite interesting for fundamental research in an electron microscope. By introducing laser into an electron microscope, the low-mag general scan can be collected by the authors' method, once area of interest is found, a detailed research can be realized by electron microscopy. [Nano Letters 16 (10), 6008-6013] On line 230, the " is probably a redundant typo.

Reviewer 3 Report

In this work, Lubcke and coauthors report a method to monitor the evolution of LIPSS by imaging a characteristic fringe pattern from the backscattered field from the formed LIPSS. 

They image the formed pattern from LIPSS fabricated on Ti and Si3N4 samples and compare their results with numerically obtained results. 

Overall the method could be useful. However, this reviewer does not believe that the paper is publishable in its current form and extensive revisions should be made. 

My concerns are as follows:

1- Abstract:

(i)The abstract is unclear. For instance, reading the abstract, I was under the impression that the fringe pattern is a well known phenomenon since it is reported in the first sentence as a fact.

(ii) I am also not sure that the term online monitoring is a proper one. If you have examples from existing literature, please provide them. Otherwise the term monitoring is sufficient to relay the intended meaning, i.e., that you are imaging the event as it happens. 

2- Introduction:

(i) The introduction is weak and certainly more information is needed. You should have provided the readers with the physical origin of LIPSS and its importance. In addition, later on you will discuss Airy desks and the fringe pattern formed on it due to LIPSS. It is thus necessary to provide sufficient background to these processes as well as their relevance in other in situ diffraction imaging methods which are quite common. 

(ii) The authors claim that the patterning laser's wavelength is longer than the LIPSS period. This is incorrect since in  the LIPSS period could be longer than the laser wavelength when the angle of incidence is not zero. 

(iii) the authors stated that observing LIPSS with the patterning laser is not possible since they are shorter than the wavelength. If my understanding is correct, then this statement is also incorrect. The formed LIPSS period could be shorter than the patterning laser wavelength yet not below its diffraction limit in most cases.

Experimental:

(i) The authors discuss the nanostructuring experiment as a part of a proton accelerator. I was for a while quite confused since i thought that the accelerator is relevant somehow to the whole fabrication process. I urge the authors to provide information only on the relevant part of the experiment, i.e., the femtosecond laser and not the particle accelerator's details which can only confuse the reader.

(ii) Why did the author use a mirror for M2 and not a beam splitter?

Computational:

The authors stated that they obtained an "Airy like beam profile". I do not think this is an accurate description. My understanding is that airy beams are a type of beam profiles that appears to curve as it travels. I think the authors meant an Airy disk profile. 

Results:

(i) what is the effect of the distance from the sample to the detector on the fringe pattern?

(ii) When examining the effect of the size of the structured area, what boundary conditions did you use for simulations?

(iii) when you considered the effect of the period, you considered the case for "600 nm, 700 nm, 800nm and 200 nm," please put them in order. Also, you what is the effect of the distance from the source on the fringe periods. 

(iv) I do not understand the point made regarding optical vs. geometric properties.  Experimentally, you could just shine the sample with low intensity pulse and see the fringe pattern backscattered from the already formed LIPSS. If it is the same, then the effect is not related to transient changes in reflectivity, etc.

Round 2

Reviewer 3 Report

The authors adequately addressed my comments and concerns.